# Therapeutic Potential of Cinnamon Oil: Chemical Composition, Pharmacological Actions, and Applications

**DOI:** 10.3390/ph17121700

**Published:** 2024-12-17

**Authors:** Jiageng Guo, Xinya Jiang, Yu Tian, Shidu Yan, Jiaojiao Liu, Jinling Xie, Fan Zhang, Chun Yao, Erwei Hao

**Affiliations:** 1Guangxi Key Laboratory of Efficacy Study on Chinese Materia Medica, Guangxi University of Chinese Medicine, Nanning 530000, China; triumph325@163.com (J.G.); jxy845923587@163.com (X.J.); 18389104087@163.com (Y.T.); yanshidu2021.gxtcmu@vip.163.com (S.Y.); 16634253325@163.com (J.L.); 13257716536@163.com (J.X.); zhangfanzf78@126.com (F.Z.); 2Guangxi Collaborative Innovation Center of Study on Functional Ingredients of Agricultural Residues, Guangxi University of Chinese Medicine, Nanning 530000, China; 3Guangxi Key Laboratory of TCM Formulas Theory and Transformation for Damp Diseases, Guangxi University of Chinese Medicine, Nanning 530000, China; 4Engineering Research Center of Innovative Drugs for Traditional Chinese Medicine and Zhuang and Yao Medicine, Ministry of Education, Guangxi University of Chinese Medicine, Nanning 530000, China

**Keywords:** cinnamon oil, pharmacological action, mechanism of action, research progress, traditional Chinese medicine

## Abstract

Cinnamon oil, an essential oil extracted from plants of the genus Cinnamomum, has been highly valued in ancient Chinese texts for its medicinal properties. This review summarizes the chemical composition, pharmacological actions, and various applications of cinnamon oil, highlighting its potential in medical and industrial fields. By systematically searching and evaluating studies from major scientific databases including Web of Science, PubMed, and ScienceDirect, we provide a comprehensive analysis of the therapeutic potential of cinnamon oil. Research indicates that cinnamon oil possesses a wide range of pharmacological activities, covering antibacterial, anti-inflammatory, anti-tumor, and hypoglycemic effects. It is currently an active ingredient in over 500 patented medicines. Cinnamon oil has demonstrated significant inhibitory effects against various pathogens comprising *Staphylococcus aureus*, *Salmonella*, and *Escherichia coli*. Its mechanisms of action include disrupting cell membranes, inhibiting ATPase activity, and preventing biofilm formation, suggesting its potential as a natural antimicrobial agent. Its anti-inflammatory properties are evidenced by its ability to suppress inflammatory markers like vascular cell adhesion molecules and macrophage colony-stimulating factors. Moreover, cinnamon oil has shown positive effects in lowering blood pressure and improving metabolism in diabetic patients by enhancing glucose uptake and increasing insulin sensitivity. The main active components of cinnamon oil include cinnamaldehyde, cinnamic acid, and eugenol, which play key roles in its pharmacological effects. Recently, the applications of cinnamon oil in industrial fields, including food preservation, cosmetics, and fragrances, have also become increasingly widespread. Despite the extensive research supporting its medicinal value, more clinical trials are needed to determine the optimal dosage, administration routes, and possible side effects of cinnamon oil. Additionally, exploring the interactions between cinnamon oil and other drugs, as well as its safety in different populations, is crucial. Considering the current increase in antibiotic resistance and the demand for sustainable and effective medical treatments, this review emphasizes the necessity for further research into the mechanisms and safety of cinnamon oil to confirm its feasibility as a basis for new drug development. In summary, as a versatile natural product, cinnamon oil holds broad application prospects and is expected to play a greater role in future medical research and clinical practice.

## 1. Introduction

Cinnamon is derived from the dried bark of several species in the *Cinnamomum* genus, including *Cinnamomum cassia*, *Cinnamomum verum*, and *Cinnamomum zeylanicum*. *C. cassia*, commonly referred to as Chinese cinnamon, is primarily sourced from China, especially the Guangxi, Guangdong, and Hainan provinces. On the other hand, *C. verum*, also known as Ceylon cinnamon, is native to Sri Lanka and parts of southern India, while *C. zeylanicum* (a historical synonym for *C. verum*) is similarly recognized for its sweeter and more delicate flavor profile [1]. Cinnamon has a long history of medicinal and culinary use. In China, the medicinal properties of *C. cassia* were documented in classical texts such as the Tang Materia Medica and Shen Nong’s Materia Medica. Meanwhile, *C. verum* has been widely used in Ayurvedic medicine and other traditional healing practices across South Asia. Globally, cinnamon is a staple spice and medicinal ingredient, and in China alone, over 500 varieties of patent medicines include cinnamon as a key component. In traditional Chinese medicine, various species of cinnamon are employed to treat numerous conditions, including erectile dysfunction, uterine coldness, cold-induced pain in the lower back and knees, kidney deficiency, respiratory issues, Yang deficiency, dizziness, eye redness, abdominal discomfort caused by cold, cold-related diarrhea, hernia, abdominal pain, and painful menstruation. As one of the largest producers of cinnamon in the world, China plays a critical role in global cinnamon production. The country accounts for a significant portion of the global cinnamon supply, making it a dominant player in the global market. The cultivation of cinnamon in China spans extensive areas, with large-scale plantations and a long-standing history of farming practices. These extensive planting areas, coupled with China’s favorable climate for growing *C. cassia*, ensure that the country remains a primary source of cinnamon for both domestic consumption and international trade [2]. The *C*. genus belongs to the *Lauraceae* family, a group of evergreen trees and shrubs commonly found in tropical and subtropical regions. The plants typically thrive in humid climates, with lush, dark green foliage and aromatic oils present in the leaves, bark, and sometimes the fruit. Cinnamon plants can be grown in various conditions, but they prefer well-drained soils and warm, humid environments. After the bark is harvested, it is peeled from the branches, then dried and curled into the familiar quill shape, or ground into powder for use in products like cinnamon oil, capsules, and culinary preparations. Additionally, modern research on cinnamon oil—derived from *C. cassia*, *C. verum*, and other species—has revealed a wide range of pharmacological effects. These include antibacterial, anti-inflammatory, antioxidant, anti-anxiety, anti-tumor, and hypoglycemic activities. Despite variations in flavor and chemical composition between the species, cinnamon oil from all types of cinnamon exhibit these beneficial health effects, making cinnamon a versatile botanical in both traditional and modern medicine [3,4,5,6,7,8,9,10,11,12,13,14,15,16,17,18,19].

Research on the antibacterial spectrum of cinnamon oil and its components has revealed varying degrees of efficacy against gram-positive bacteria such as *S. aureus* and *Candida albicans*, as well as gram-negative bacteria including *Salmonella*, *E. coli*, and *Pseudomonas aeruginosa*, studies on its antifungal properties have also shown promising results, particularly against *Candida* species (*Candida albicans*, *Candida glabrata*, *Candida tropicalis*) and other pathogenic fungi such as *Aspergillus niger*, *Aspergillus fumigatus*, and *Cryptococcus neoformans* [20,21,22,23,24,25]. Furthermore, research on the anti-inflammatory properties of cinnamon oil indicates that it exerts immunomodulatory effects by inhibiting the production of key inflammatory mediators [9,10,17,26]. In terms of metabolic health, cinnamon oil has been shown to help regulate blood glucose levels and lipid metabolism, with studies demonstrating its ability to lower triglycerides, low-density lipoprotein (LDL), and total cholesterol in diabetic patients [10,27,28]. Cinnamon oil is also recognized for its therapeutic effects on gastrointestinal disorders, which are thought to be mediated by its active compounds acting on the gastrointestinal tract [29,30,31]. Additionally, due to its potent natural antioxidant properties, cinnamon oil is widely used in the food industry to prevent oxidation in products, and it has found applications in medicine for its role in neutralizing free radicals [22,32]. This review discusses the chemical composition, pharmacological effects, and diverse applications of cinnamon oil, aiming to provide a comprehensive reference for its future development and utilization across both medical and industrial fields (Figure 1).

### Review Methodology

To comprehensively review the pharmacological effects of cinnamon oil from various species, a systematic search was conducted across major scientific databases, including Medline, PubMed, ScienceDirect, and Scopus, covering the publication period from 1 January 2018 to 31 July 2024. In addition, a manual search was performed to identify relevant articles that may have been missed in the database queries. The literature retrieval process was designed to encompass a broad range of studies discussing both the molecular mechanisms and therapeutic potential of cinnamon oil from multiple *C*. species, including but not limited to *C*. *cassia*, *C*. *verum*, *C*. *zeylanicum*, and *C*. *burmannii*. The search strategy employed specific keywords such as “Cinnamon oil”, “*C*. *cassia*”, “*C*. *verum*”, “*C*. *zeylanicum*”, “mechanisms of action”, and “pharmacological effects”, in combination with relevant synonyms and related terms. Boolean operators (AND/OR) were utilized to ensure comprehensive coverage of all relevant studies on the pharmacological effects of these cinnamon oils. The inclusion criteria for this review were predefined to consider peer-reviewed research articles, review papers, and clinical trial reports published in English. Studies addressing both in vitro and in vivo experiments were included to provide a complete overview of the pharmacological profile of cinnamon oils. Articles not directly related to the pharmacological actions of cinnamon oil or focused solely on its chemical synthesis without addressing biological effects were excluded.

## 2. Chemical Composition

Cinnamon oil, derived from various species within the Cinnamomum genus, has a rich chemical composition. The extraction methods for cinnamon’s active compounds have evolved significantly over time, allowing researchers to analyze and harness its therapeutic potential from different plant species, including *C. cassia*, *C. verum*, *C. zeylanicum*, and *C. burmannii*. The two most commonly used techniques for extracting cinnamon oil are steam distillation and supercritical CO_2_ extraction [33,34]. Gas chromatography-mass spectrometry (GC-MS) is the primary method used to analyze and identify the chemical compounds present in cinnamon oil, distinguishing between volatile and non-volatile components. Cinnamon oil can be extracted from various parts of the plant, including the bark, leaves, and branches, with each part contributing slightly different chemical profiles. The volatile oils, rich in bioactive compounds, are considered the primary active components of cinnamon oil and are utilized in various sectors such as food, medicine, and catalysis [35]. In addition to volatile oils, cinnamon also contains coumarin, an important bioactive compound [36]. Coumarin exists both in essential oils and in free form. It has a variety of biological activities such as anticoagulant, anti-inflammatory, and antioxidant, but may be hepatotoxic at high concentrations. Therefore, the content of coumarin needs to meet the relevant pharmacopoeia standards to ensure its safe use. According to the 2020 edition of the Chinese Pharmacopoeia [37], the coumarin content in *C. cassia* essential oil shall not exceed 0.1%. This standard ensures the safe use of essential oils and avoids the potential toxicity caused by excessive coumarin.

Cinnamon essential oil primarily consists of cinnamaldehyde, eugenol, and several other bioactive compounds. There are significant differences in the chemical composition of different species of cinnamon in the genus Cinnamomum, which are mainly reflected in the content and composition of the main bioactive compounds. For example, the content of cinnamaldehyde in *C. cassia* and *C. verum* is higher, while the content of eugenol in *C. verum* is significantly higher than that in other species. This chemical diversity directly affects the biological activities of cinnamon essential oil, such as antibacterial, anti-inflammatory, and anti-cancer effects (Table 1). In *C. cassia* and *C. verum*, cinnamaldehyde typically ranges from 60% to 90%, serving as the main component responsible for its antibacterial, anti-inflammatory, and anti-cancer properties [38,39,40]. Cinnamaldehyde, the main compound in cinnamon essential oil, is an aromatic aldehyde with the chemical formula C_9_H_8_O. It consists of a benzene ring (C_6_H_5_) attached to a propenyl group (-CH=CH_2_) and an aldehyde functional group (-CHO) at the alpha position. Its molecular structure gives cinnamaldehyde its characteristic spicy, sweet aroma and accounts for its antimicrobial and therapeutic properties. The compound has strong antioxidant activity, making it useful for medicinal purposes, and is also known for its ability to inhibit the growth of bacteria and fungi. Cinnamaldehyde is volatile, soluble in ethanol and ether, and unstable at high temperatures, which can cause its degradation. Eugenol, a key compound found predominantly in *C. verum* (38.42% in bark [41]), though this can vary based on extraction methods and cultivation factors, contributes to the oil’s anti-inflammatory and analgesic effects. Other significant constituents include cinnamyl acetate (5% to 10%), linalool (1% to 5%), β-caryophyllene (1% to 3%), benzyl benzoate, clovene, eugenol acetate, and various inorganic elements, each present in smaller concentrations [19,42]. The composition of cinnamon oil varies slightly between species, reflecting the different geographical and environmental conditions where they are grown. The amounts of essential oil and volatile compounds in cinnamon essential oil also depend on the extraction conditions, such as the temperature, duration, and method used. The most common method of extracting cinnamon essential oil is aqueous extraction, which involves distilling the plant material with distilled water. This method is widely used because it is simple and cost-effective while retaining much of the oil’s bioactive properties. Cinnamon oil is widely applied across several industries. The oil’s versatility and its broad range of health benefits underscore its significant role in various sectors, from medicine to everyday consumer goods.

## 3. Antibacterial Effect

China has significant reserves of *C. cassia*, making it a leading producer of cinnamon worldwide. Traditionally, cinnamon oil, extracted from this spice, has been widely used for its antibacterial and antiseptic properties. Research has shown that cinnamon oil is a versatile antibacterial agent, effective against both bacteria and fungi. Notably, cinnamaldehyde, one of the primary active components in cinnamon oil, has been approved by the U.S. Food and Drug Administration as a safe food additive [43]. While most research on the antibacterial properties of cinnamon oil has focused on its contact-based bacteriostatic effects, recent studies suggest that the antibacterial action of cinnamon oil in its vapor phase may be even more potent. Essential oils in vapor form, including cinnamon oil, have been shown to inhibit a variety of pathogenic bacteria at lower concentrations, offering potential cost savings in medical and industrial applications [44]. This highlights the importance of further investigating the specific mechanisms by which cinnamon oil, both in contact and vapor forms, combats pathogenic bacteria. Such research is crucial for enhancing its application in human health protection.

### 3.1. Anti-S. aureus

*S. aureus* is a common cause of foodborne illnesses and is widely present in various environments. In China, it accounts for 25% of bacterial food poisoning cases and poses significant health risks due to certain strains producing staphylococcal enterotoxin [45]. Antibiotic resistance in *S. aureus* is on the rise, and searching for new antibacterial agents is urgent [46]. Cinnamon oil, a natural plant extract, is known for its antibacterial activity, particularly against *S. aureus*.

Zhang Yunbin et al. [47] evaluated the antibacterial properties of cinnamon oil, specifically from *C. cassia*, against *S. aureus*. They directly cultured the bacteria in a culture medium corresponding to a series of cinnamon oil concentration and they determined a minimum inhibitory concentration (MIC) of 1.0 mg/mL, a minimum bactericidal concentration (MBC) of 2.0 mg/mL, and an inhibition zone (DIZ) of 28.7 mm. Gas chromatography-mass spectrometry (GC-MS) analysis identified cinnamaldehyde as the predominant component, accounting for 92.40% of the oil’s composition. Microscopic examination at the MIC level revealed significant morphological changes in the bacterial cells, characterized by increased permeability and compromised membrane integrity. At the MBC level, complete cell destruction was observed. Conductivity measurements indicated enhanced membrane permeability and leakage, with protein and nucleic acid concentrations in the suspension, increasing proportionally with cinnamon oil concentration, ultimately leading to cell death. Additionally, membrane potential measurements demonstrated a 3–5 fold reduction in bacterial metabolic activity, indicating the oil’s potent antibacterial effects [48].

Celso Vataru Nakamura et al. [49] investigated the antibacterial properties of cinnamon oil from *C. verum* against *S. aureus*. Their findings indicated that cinnamon oil exhibited significant antibacterial activity with a minimum inhibitory concentration (MIC) of 1.6 mg/mL and a minimum bactericidal concentration (MBC) of 1.6 mg/mL, which were identical to its MIC values. In contrast, cinnamaldehyde demonstrated similar MIC values of 1.6 mg/mL, but required a higher concentration of 6.4 mg/mL to achieve bactericidal effects (MBC). The study further revealed that cinnamon oil disrupts the cell wall of bacteria, compromising its protective structure and ultimately leading to bacterial cell death. Notably, the predominant compound in the cinnamon oil was found to be cis-cinnamaldehyde, comprising 59.7% of its composition.

Methicillin-resistant *S. aureus* (MRSA) is associated with severe skin and respiratory infections [50] and is a major cause of hospital-acquired infections due to its high drug resistance [51]. Although vancomycin, linezolid, daptomycin, and tigecycline are FDA-approved for MRSA treatment [52], alternative antibiotics are still being sought.

Mustofa Helmi Effendi et al. [53] investigated the antibacterial effects of cinnamon oil, extracted from *C. burmannii* via steam distillation, against methicillin-resistant *S. aureus* (MRSA) using the disk diffusion method. The study prepared cinnamon oil in varying concentrations of 1%, 2%, 4%, and 8% and tested its efficacy on five MRSA isolates collected from cow’s milk. The results showed that 4% cinnamon oil effectively inhibited all tested MRSA strains, with inhibition zones exceeding 8 mm, while 8% cinnamon oil produced an average inhibition zone diameter of 20 mm. Several studies and reviews have attempted to elucidate the mechanism of action, and their results suggest that cinnamon oil damages bacterial cell membranes [54,55,56,57], leading to increased membrane permeability and ultimately cell death. High concentrations of cinnamon oil may result in significant morphological changes, including swelling and rupture of bacterial cells. In addition, the release of intracellular potassium ions (K+) indicates membrane damage, and the lipophilicity of cinnamon oil facilitates its penetration into the bacterial membrane, altering its properties and enhancing permeability.

In summary, cinnamon oil exhibits antimicrobial activity against *S. aureus* and MRSA by disrupting cell membranes, inhibiting ATPase function, and hindering biofilm formation. These findings suggest its potential as a therapeutic agent for treating *S. aureus* infections.

### 3.2. Anti-Salmonella

Food safety and health are increasingly important concerns. According to the World Health Organization, *S*. is a prevalent zoonotic pathogen [58], found globally and significantly impacting public health and the economy [59]. *S*., a gram-negative bacterium in the Enterobacteriaceae family, has flagella, lacks spores, and appears red after staining [60]. *S. Enteritidis*, a foodborne pathogen, invades the intestinal tract upon consuming contaminated food [61]. It disrupts intestinal barrier function, causing dysbiosis and leading to liver injury marked by inflammation, oxidative stress, hepatocyte apoptosis, and severe congestion [62]. Currently, antibiotics like chloramphenicol and levofloxacin are used to combat *S*. infections [63,64]. However, antibiotic overuse has led to resistant strains, necessitating alternative antibacterial agents.

Zhao Yuanyuan et al. [22] evaluated the antibacterial effect of cinnamon oil (CO) against *S. Enteritidis*. In the experiment, CO was extracted by steam distillation from cinnamon bark purchased from Guangdong, China. Using an in vitro model, bacterial suspensions in the logarithmic growth phase were treated with CO at 1/2 MIC (0.4 μL/mL), MIC (0.8 μL/mL), and 2 MIC concentrations, and a control group was set up. However, the study did not specify whether a positive control (such as a standard antibiotic) was used for comparison, which may limit the interpretability of the results. They used fluorescence microscopy to observe the levels of reactive oxygen species (ROS) in CO-treated bacteria and measured malondialdehyde (MDA) content and the activities of antioxidant enzymes (such as superoxide dismutase SOD, catalase CAT, and peroxidase POD). The results showed that CO treatment led to a significant increase in the level of ROS in bacterial cells, showing a concentration-dependent manner. This excessive accumulation of ROS exceeded the bacteria’s own antioxidant defense capacity, resulting in irreversible damage to proteins and nucleic acids. In addition, the decreased activities of protective enzymes further indicated the weakening of bacterial antioxidant capacity. Exposure to antibacterial agents increases the levels of reactive oxygen species (ROS) in microorganisms, leading to a decrease in proteins and other biological macromolecules [65,66]. While protective enzymes such as superoxide dismutase (SOD), catalase (CAT), and peroxidase (POD) can neutralize ROS [67], excessive ROS can overwhelm the bacterial antioxidant defenses, causing irreversible damage to proteins and nucleic acids [68].

Some other studies have elucidated possible mechanisms of action [69,70,71,72]: higher concentrations of CO increase the production of ROS in *S. Enteritidis*, leading to severe oxidative damage and antimicrobial effects. As ROS levels increase, cellular lipid peroxidation also increases, thereby enhancing the antimicrobial effect. In addition, the effect of CO on the gene expression of outer membrane proteins (Omps) of *S. Enteritidis* was also studied. Omps (including OmpA, OmpF, OmpW, and OmpX) play a vital role in bacterial metabolism, drug entry regulation, cell shape maintenance, and the synthesis of related metabolites. These studies show that CO increases the permeability of these proteins, disrupts bacterial metabolism, and indicates its potential as a natural antibiotic alternative.

### 3.3. Anti-E. coli

*E. coli* is a Gram-negative rod-shaped bacterium in the *Enterobacteriaceae* family [73]. Certain strains cause various infections, including intra-abdominal infections, urinary tract infections (UTIs), pulmonary infections, skin infections, soft tissue infections, diarrhea, and bacteremia by evading host immune defenses and producing toxins [74,75]. Overuse of antibiotics has reduced the effectiveness of conventional treatments, leading to multi-drug resistant *E. coli* and posing a significant global public health threat [76]. Therefore, finding alternative treatments for *E. coli* infections is crucial. Cinnamon oil, known for its broad-spectrum antibacterial properties, has emerged as a potential therapeutic agent against *E. coli.*

Liu Xiaoyu et al. [47] also investigated the antibacterial properties of CO against *E. coli.* The researchers determined the minimum inhibitory concentration (MIC) of CO against *E. coli* to be 1.0 mg/mL and the minimum bactericidal concentration (MBC) to be 4.0 mg/mL using broth microdilution methods. The diameter of the inhibition zone (DIZ) was measured via the agar diffusion method, resulting in a DIZ of 19.2 mm. While these findings demonstrate CO’s antibacterial activity, the study did not specify the use of positive controls (such as standard antibiotics) for comparison, which limits the assessment of CO’s relative efficacy. Conductivity measurements indicated that higher doses of CO enhanced cell membrane permeability, resulting in the leakage of intracellular electrolytes and macromolecules. This leakage implies a loss of membrane integrity, which is critical for maintaining cellular homeostasis and function. The study also assessed the bacterial membrane potential (MP) using the mean fluorescence intensity (MFI) of Rhodamine 123 dye. Results showed significant decreases in MFI values at MIC (a 41.49% reduction) and MBC (an 80.77% reduction) levels compared to the control group. The reduction in membrane potential suggests depolarization of the bacterial cell membrane, which disrupts essential metabolic activities and can ultimately lead to cell death. To explore the antibacterial mechanism, scanning electron microscopy (SEM) was employed. At the MIC level, SEM images revealed significant morphological alterations in *E. coli* cells, including disrupted membrane integrity and increased permeability. At higher concentrations corresponding to the MBC, cells exhibited wrinkled and irregular surfaces, with some cells showing swelling and rupture. These observations suggest that CO compromises the bacterial cell wall and membrane, leading to structural damage [77].

*Avian pathogenic E. coli* (APEC) causes significant financial losses in poultry production due to yolk sac infections, septicemia, air sac inflammation, and lesions in internal organs [78]. Gaia Casalino et al. [79] investigated the antibacterial effects of cinnamon oil (CO) against pathogenic *E. coli* strains isolated from poultry affected by colibacillosis. The study utilized a collection of 117 *E. coli* strains from laying hens, broilers, and turkeys, and assessed the minimum inhibitory concentrations (MICs) of CO using in vitro assays. The results demonstrated that CO exhibited significant antibacterial activity, with MIC values ranging from 0.2 to 0.8 µL/mL, indicating noteworthy efficacy against all tested strains. Cinnamaldehyde, identified as the primary active metabolite in CO, plays a crucial role in its antibacterial action. Studies have shown that cinnamaldehyde prevents pathogenic *E. coli* from adhering to surfaces [80], which is essential for infection establishment. It inhibits bacterial growth by prolonging the lag phase, increasing cell membrane permeability, and inducing oxidative damage that leads to membrane collapse and leakage of cellular contents. These mechanisms disrupt bacterial metabolism and viability. Moreover, the study highlighted the potential synergistic effects of CO when combined with conventional antibiotics. Specifically, the combination of CO with chloramphenicol achieved enhanced antibacterial effects at lower concentrations [81], suggesting a synergistic action. This implies that CO could potentiate antibiotic treatments against bacterial infections, potentially allowing for reduced antibiotic dosages and mitigating the development of antibiotic resistance.

In summary, cinnamon oil has demonstrated efficacy in treating illnesses caused by pathogenic *E. coli*, offering promise as a contender for innovative infection-combating medications. It provides fresh perspectives and a theoretical foundation for developing natural antimicrobials or gut regulators.

### 3.4. Anti-P. aeruginosa

*P. aeruginosa* (PA) is a motile, aerobic bacillus with unipolar flagella [82]. It is a common pathogen in hospital settings, causing infections in various systems, including the digestive and respiratory systems. The overuse of antibiotics has led to the emergence of multi-drug-resistant and pan-drug-resistant PA strains, complicating clinical treatment [83]. In burn patients, wound infections, often complicated by PA, can lead to chronic wounds if not properly treated, highlighting the need for effective infection management [84]. Carbapenem antibiotics are typically reserved as a last resort for PA infections, but resistance to these drugs is increasing. Consequently, researchers are exploring natural products to mitigate the side effects of excessive antibiotic use and to develop new therapeutic options.

E. R. Elcocks et al. [84] investigated the antibacterial properties of CO against *P. aeruginosa* using both disk diffusion and broth microdilution methods. The CO was analyzed by gas chromatography-mass spectrometry (GC-MS), identifying cinnamaldehyde as the primary metabolite, comprising approximately 85.3% of the oil’s composition. In the disk diffusion assays, CO exhibited inhibition zones ranging from 16.7 mm to 36.2 mm, significantly larger than those produced by the gentamicin control (*p* < 0.05), indicating strong antibacterial activity. The minimum inhibitory concentration (MIC) and minimum bactericidal concentration (MBC) of CO against *P. aeruginosa* were determined to be ≤1.0% (*v*/*v*) and ≤0.25% (*v*/*v*), respectively, using the broth microdilution method. Notably, CO demonstrated inhibitory effects at concentrations as low as 0.125% (*v*/*v*). To elucidate the antibacterial mechanism, metabolic activity was assessed using a triphenyltetrazolium chloride (TTC) assay. In microplate wells containing live *P. aeruginosa*, a color change from yellow to red indicated metabolic activity due to the reduction of TTC by viable cells. Wells treated with CO concentrations of 1% (*v*/*v*) or higher showed no color change within 2 min, suggesting rapid cessation of metabolic activity and bacterial viability. Scanning electron microscopy (SEM) provided visual evidence of CO’s impact on bacterial morphology. After 10 min of exposure to 2% (*v*/*v*) CO, *P. aeruginosa* cells appeared collapsed, with fewer surface particles, indicating significant damage to the cell envelope. The study also measured the release of nucleic acids and proteins from *P. aeruginosa* following CO treatment. Elevated absorbance readings at 260 nm and 280 nm, corresponding to nucleic acids and proteins, respectively, indicated increased leakage of intracellular contents. This suggests that CO disrupts the bacterial cell membrane integrity, leading to cell lysis and death.

A study supported the antibacterial effects of cinnamon oil (CO) against *P. aeruginosa* [85]. Gas chromatography-mass spectrometry (GC-MS) analysis confirmed cinnamaldehyde as the major metabolite, constituting 99.54% of the oil. The researchers tested CO on 136 clinical isolates of *P. aeruginosa* using the disk diffusion method, applying 15 μL of CO per disk. The results demonstrated significant antibacterial activity, with inhibition zones ranging from 16 mm to 35 mm, and only one isolate showing resistance. The minimum inhibitory concentration (MIC) against *P. aeruginosa* ATCC 27853 was determined to be 0.0019 mL/mL, with an inhibition zone diameter of 21 mm.

At the same time, some studies have used cinnamon oil in combination with other materials, showing more promising antibacterial activity. In a study assessing the antibacterial and wound healing properties of cinnamon oil (CO), aloe vera (AV) gel, and their combination (CO + AV) [86], researchers evaluated their effects against *P. aeruginosa* ATCC 27853 and *S. aureus* ATCC 25923. Using the disc diffusion method, they found that the combination exhibited greater antibacterial activity than either substance alone, with a more pronounced inhibitory effect on *P. aeruginosa*. Specifically, the zone of inhibition for CO + AV against *P. aeruginosa* was 25 mm, compared to 15 mm against *S. aureus*. The minimum inhibitory concentration (MIC) of the combination was also lower for *P. aeruginosa* (100 µg/mL) than for *S. aureus* (200 µg/mL), indicating higher effectiveness against *P. aeruginosa*. In an in vivo rat excision wound model infected with these bacteria, treatment with CO + AV significantly enhanced wound healing, demonstrated by increased wound contraction, higher hydroxyproline content (indicative of collagen synthesis), and improved antioxidant activity. Histopathological analysis showed better tissue regeneration in the combined treatment group.

In summary, cinnamon oil demonstrates significant bacteriostatic and bactericidal effects against *P. aeruginosa* at low concentrations and within short exposure times. This suggests its potential as an alternative to traditional antibacterial medications. The antibacterial mechanism involves disruption of the bacterial cell membrane, leading to cell leakage. Based on these findings, cinnamon oil shows promise as an effective antibacterial agent for use in cosmetics, medicine, and the food industry.

### 3.5. Anti-Candida

Candidiasis is one of the most common invasive fungal infections [87]. *C. albicans*, a common type of invasive *Candida*, is typically found on the mucosal linings and in the genitourinary tract. It can colonize the intestines and cause both superficial and systemic infections [88]. Under normal immune function, it exists harmlessly with the host, but when the immune system is weakened, *C. albicans* can become pathogenic, entering the bloodstream and causing serious systemic infections [89].Three main classes of antifungal medications are used to treat invasive fungal infections: echinocandins, azoles, and polyenes. Despite these treatment options, resistance to traditional antifungal drugs is increasing, necessitating new therapeutic agents [90]. Cinnamon oil, known for its antibacterial properties, is being studied for its potential role in combating *C*.

Felipe Joia et al. [91] investigated the antifungal and antibiofilm activities of cinnamon oil (CO) against *C. albicans* using in vitro models. The study determined the minimum inhibitory concentration (MIC) and minimum fungicidal concentration (MFC) of CO following the Clinical and Laboratory Standards Institute (CLSI) microdilution method. The MIC was found to be 1.0 mg/mL, and the MFC was 2.0 mg/mL, indicating that CO effectively eradicated *C. albicans* within six hours. Furthermore, the minimum biofilm inhibitory concentration (MBIC) of CO was less than 0.2 mg/mL, demonstrating significant efficacy against biofilm-forming Candida cells. Scanning electron microscopy (SEM) images revealed considerable cell wall damage and cell shrinkage in treated samples, while transmission electron microscopy (TEM) showed intracellular vacuolation and impairment of cell wall integrity. These morphological changes suggest that CO disrupts the fungal cell membrane and interferes with cellular functions. Importantly, cytotoxicity assays indicated that CO concentrations up to 1000 µg/mL did not adversely affect human keratinocyte (HaCaT) cells, suggesting that CO is a viable and safe treatment option for *Candida* infections.

Veilleux et al. [92] investigated the antifungal and antibiofilm activities of cinnamon bark essential oil and a cinnamon extract against *C. albicans*. The study determined the minimum inhibitory concentration (MIC) and minimum fungicidal concentration (MFC) of the cinnamon oil, finding MIC values ranging from 0.039% to 0.078% (*v*/*v*) and demonstrating fungicidal activity with an MFC/MIC ratio ≤ 4. The cinnamon oil was shown to permeabilize the cell membrane of *C. albicans*, leading to cell death, as evidenced by increased uptake of SYTOX Green dye. Moreover, cinnamon oil significantly inhibited biofilm formation by *C. albicans* at sub-inhibitory concentrations. Cinnamon oil at 0.0049% (*v*/*v*) reduced biofilm formation by 86%, while cinnamon oil at 62.5 μg/mL reduced it by 91%. Scanning electron microscopy revealed that cinnamon oil disrupted the structural integrity of the biofilms and prevented hyphae formation. Importantly, cinnamon oil exhibited anti-inflammatory properties by reducing the secretion of pro-inflammatory cytokines IL-6 and IL-8 from human oral epithelial cells stimulated with TNF-α. At a concentration of 62.5 μg/mL, cinnamon oil reduced IL-6 and IL-8 secretion by 29% and 57%, respectively. Cytotoxicity assays indicated that both cinnamon oil were non-toxic to oral epithelial cells at effective concentrations, suggesting their potential safety for therapeutic use.

In summary, cinnamon oil demonstrates anti-*Candida* properties by impacting biofilms and showing no harmful effects on human cells, suggesting its potential as a promising alternative for managing *C. albicans* infections (Figure 2).

## 4. Antibacterial *lipopolysaccharide* (LPS)-Induced Inflammation

Inflammation is a complex immune response that protects the body from pathogens, toxins, and environmental stressors [93]. Failure to resolve inflammation or prolonged stimulation can lead to chronic activation of immune cells, resulting in health issues and tissue damage [94]. While nonsteroidal anti-inflammatory drugs (NSAIDs) and glucocorticoids are commonly used to treat inflammation, they have significant side effects and are not suitable for long-term use. Consequently, there is a need for new, safer anti-inflammatory treatments. Cinnamon oil, a natural product, has shown promise due to its anti-inflammatory properties. Bacterial *lipopolysaccharide* (LPS), primarily produced by Gram-negative bacteria, is a key factor in triggering inflammation and disrupting normal epithelial function, posing a significant health threat [95]. LPS-induced inflammation is associated with diseases such as bronchitis, pneumonia, and hepatitis [96]. Essential oils are frequently used to prevent and treat various ailments, including acute lung injury and asthma [97]. Research indicates that cinnamon oil, derived from cinnamon, has remarkable anti-inflammatory properties [98].

Zhao Chen et al. [6] conducted a study to assess the anti-inflammatory effects of CO in LPS-induced mice models. Mice were pre-treated with CO at doses of 500 mg/kg, 750 mg/kg, and 1000 mg/kg, administered orally. The study measured various physiological and biochemical parameters, including changes in body weight, visceral edema coefficients of the liver and kidneys, and levels of antioxidant enzymes such as superoxide dismutase (SOD) and glutathione peroxidase (GSH-Px). Additionally, they evaluated markers of oxidative stress and inflammation, including malondialdehyde (MDA), nitric oxide (NO), interleukins IL-6 and IL-10, tumor necrosis factor-alpha (TNF-α), alanine aminotransferase (ALT), toll-like receptor 4 (TLR4), nuclear factor kappa B (NF-κB), inducible nitric oxide synthase (iNOS), and manganese superoxide dismutase (Mn-SOD). Histopathological examination of liver tissue was performed using hematoxylin and eosin (H&E) staining to assess tissue damage. The results indicated that CO effectively scavenged free radicals, delayed the depletion of SOD and GSH-Px activities, and reduced MDA levels, thereby mitigating oxidative stress induced by LPS. CO treatment also decreased ALT levels, suggesting hepatoprotective effects by alleviating LPS-induced liver damage. Furthermore, Ref. [54] CO reduced the secretion of pro-inflammatory cytokine TNF-α, inhibited the activation of the NF-κB pathway, decreased the expression of TLR4 and iNOS, and increased Mn-SOD levels. These effects collectively contributed to the reduction of liver and kidney edema in LPS-induced mice.

A recent study also helps to support the findings [13], the anti-inflammatory effects of cinnamon oil were evaluated in a *lipopolysaccharide* (LPS)-induced inflammation model in mice. Mice were orally administered cinnamon oil at doses of 15 mg/kg/day, 30 mg/kg/day, and 60 mg/kg/day. The study measured various physiological and biochemical parameters to assess the anti-inflammatory potential of cinnamon oil. Treatment with cinnamon oil significantly reduced the levels of pro-inflammatory cytokines such as tumor necrosis factor-alpha (TNF-α), interleukin-6 (IL-6), and interleukin-1 beta (IL-1β). Additionally, it increased the levels of the anti-inflammatory cytokine interleukin-10 (IL-10). The oil also inhibited the activation of the Toll-like receptor 4 (TLR4)/myeloid differentiation factor 88 (MyD88)/nuclear factor kappa B (NF-κB) signaling pathway, which plays a crucial role in mediating inflammatory responses. This was evidenced by decreased protein and mRNA expression levels of TLR4, MyD88, and phosphorylated NF-κB p65 in tissue samples. Furthermore, cinnamon oil modulated macrophage polarization by inhibiting the pro-inflammatory M1 phenotype and promoting the anti-inflammatory M2 phenotype. These findings suggest that cinnamon oil exerts significant anti-inflammatory effects by modulating key inflammatory signaling pathways and cytokine production, highlighting its potential as a therapeutic agent for treating inflammatory conditions.

In conclusion, cinnamon oil demonstrates substantial anti-inflammatory properties in LPS-induced inflammation, suggesting its protective role and potential to prevent associated conditions.

## 5. Antioxidant Properties of Chemical Components of Cinnamon Oil

In recent years, extensive research has been conducted on the phytochemical composition of essential oils, revealing their potential roles in various health applications such as diabetes prevention, inflammation reduction, and combating chronic diseases. These conditions are often linked to oxidative damage caused by free radicals [99]. Methods for assessing the phytochemical properties of essential oils often rely on chemical analysis or cell biology evaluations [100]. However, it is crucial to note that these chemical assays, such as the DPPH or ABTS assay, are analytical tools with no direct pharmacological relevance. Therefore, findings from such assays should be interpreted cautiously and cannot be directly translated into therapeutic benefits.

Chen Xiaohua et al. [32] conducted a comprehensive study to evaluate the antioxidant capacity of cinnamon oil using multiple assays, including the DPPH free radical scavenging method, a fish oil-in-water emulsion system, and a cellular antioxidant activity assay in red blood cells (CAA-RBC). In the DPPH assay, cinnamon oil exhibited strong free radical scavenging activity, with an EC 50 value of 0.03 mg/mL, comparable to the positive control, Trolox. This indicates a high antioxidant potential in chemical systems. When tested in the fish oil-in-water emulsion system, which models lipid oxidation in biological systems, the antioxidant EC 50 values of cinnamon oil ranged from 20.8 to 32.4 mg/mL. These values suggest moderate antioxidant activity in inhibiting lipid oxidation, showing that the oil is less effective in this complex system compared to the DPPH assay. In the cellular antioxidant activity assay using red blood cells, cinnamon oil demonstrated an EC 50 value of 0.47 mg/mL, which is higher than that of Trolox (EC 50 ≥ 0.25 mg/mL). This indicates that while cinnamon oil has antioxidant effects in cellular systems, its potency is lower than that of Trolox in this context.

Studies have revealed the remarkable antioxidant properties shared by CO and nanoparticles. Researchers explored the antioxidant potential of CO when combined with metal oxide nanoparticles such as *titanium dioxide* (*TiO_2_*), *aluminum oxide* (*Al_2_O_3_*), and *calcium carbonate* (*CaCO_3_*) [101]. The nanoparticles were modified by impregnating them with CO, resulting in CO-loaded nanoparticles. These modified nanoparticles were thoroughly characterized using techniques like Fourier-transform infrared spectroscopy (FTIR), X-ray diffraction (XRD), thermogravimetric analysis (TGA), and X-ray photoelectron spectroscopy (XPS) to confirm the successful incorporation of CO and to understand the interactions between CO and the nanoparticles. The antioxidant activity of the CO-loaded nanoparticles was evaluated using the DPPH radical scavenging assay. The results demonstrated that *Al*_2_*O*_3_ nanoparticles loaded with CO exhibited the highest antioxidant activity, achieving a 55% inhibition of DPPH radicals. This was followed by *CaCO_3_*-CO nanoparticles with a 35% inhibition and *TiO_2_*-CO nanoparticles with a 28% inhibition. In comparison, pure CO showed an 80% inhibition under the same conditions. The pure nanoparticles without CO showed minimal antioxidant activity, indicating that the antioxidant properties were primarily due to the CO. The enhanced antioxidant activity, especially in the *Al_2_O_3_*-CO system, suggests a strong interaction between the CO molecules and the surface of the *Al_2_O_3_* nanoparticles. XPS analysis confirmed the formation of chemical bonds between CO and the nanoparticles, particularly through oxygen atoms, which likely contributed to the improved antioxidant effect. This combination of cinnamon oil with metal oxide nanoparticles effectively transferred the antioxidant potential of CO to the nanoparticles, creating materials with enhanced antioxidant properties.

These findings indicate that cinnamon oil possesses antioxidant properties in vitro. However, the relevance of these results to human health requires further investigation. While chemical assays demonstrate the potential antioxidant capacity of cinnamon oil, they do not establish therapeutic efficacy. Therefore, additional studies, particularly clinical trials, are necessary to validate the health benefits of cinnamon oil as a natural antioxidant. In conclusion, although cinnamon oil has shown notable antioxidant properties in chemical assays, it is important to distinguish between these preliminary findings and their pharmacological relevance. This research lays the groundwork for exploring essential oils as natural antioxidants, but their application in enhancing human health remains to be conclusively proven through rigorous clinical studies.

## 6. Anti-Anxiety

Anxiety is a common mental disorder characterized by excessive tension and fear [102], including subtypes such as generalized anxiety disorder, obsessive-compulsive disorder, and post-traumatic stress disorder [103]. The main treatments for anxiety involve Selective Serotonin Reuptake Inhibitors (SSRIs) like *paroxetine*, *sertraline*, *citalopram*, *escitalopram*, *fluvoxamine*, and *fluoxetine* [104]. However, long-term use of these medications can lead to negative side effects, such as bleeding, digestive problems, hyponatremia, excessive sweating, emotional blunting, increased risk of suicide [105], and tachyphylaxis [106]. This underscores the urgent need for alternative treatments that effectively manage anxiety with fewer adverse reactions.

A study by Yang In-Jun et al. [8] investigated the anxiolytic effects of cinnamon oil (CO) inhalation on mouse behavior using various behavioral tests. Mice were exposed to 5% and 10% concentrations of CO and assessed using the Elevated Plus Maze (EPM), a standard test for evaluating anxiety-like behavior in rodents. The results showed that mice inhaling CO spent significantly more time in the open arms and less time in the closed arms compared to control mice, indicating a reduction in anxiety levels. Importantly, locomotor activity measured by the Open Field Test (OFT) showed no significant changes, suggesting the anxiolytic effects were not due to increased general activity. In a social interaction test, mice exposed to CO demonstrated increased social behaviors, such as a higher total contact number, especially at the 5% concentration. To further explore the mechanism behind CO’s anxiolytic properties, the gene expression profile of the mouse hippocampus was analyzed. Important genes such as *Dcc*, *Egr2*, and *Fos*, which are involved in anxiety-related processes, were examined. *Dcc* is associated with axon guidance and neuronal cell death and is linked to anxiety-like behavior [107,108]. *Egr2* is a transcription factor related to stress response and mental disorders [109]. *Fos* is a marker of neural activity, and the absence of *c-fos* is linked to reduced anxiety-like behavior in mice [110,111]. The down-regulation of *Dcc*, *Egr2*, and *Fos* genes in the hippocampus of mice treated with CO suggests that it may exert its anxiolytic effects through the modulation of these genes. Additionally, the experiment utilized a zebrafish model to assess the anti-anxiety effects of cinnamon oil. The study found that cinnamaldehyde exposure could reverse changes in brain electrical activity caused by MK-801 [112], particularly in delta and beta wave activities measured by electroencephalogram (EEG). These findings suggest that inhalation of cinnamon oil can reduce anxiety-like behavior, potentially through modulation of serotonergic and neuroinflammatory pathways, highlighting (E)-cinnamaldehyde as a key active compound.

Another study explored the antidepressant and anxiolytic effects of cinnamon oil administered intraperitoneally in mice [113]. The essential oil was given at doses of 0.5, 1, and 2 mg/kg over a sub-acute period of 14 days. Behavioral assessments using the Forced Swim Test (FST) and Tail Suspension Test (TST) showed that cinnamon oil significantly reduced immobility time, indicating antidepressant-like effects comparable to those of standard drugs *desipramine* and *fluoxetine*. In the Elevated Plus Maze (EPM) test, a dose of 2 mg/kg of the essential oil increased both the percentage of time spent and entries into the open arms, demonstrating anxiolytic properties without affecting locomotor activity in the Open Field Test. Gas chromatography–mass spectrometry (GC–MS) analysis revealed that trans-cinnamaldehyde was the main component of the essential oil, accounting for 87.32% of its composition. These findings suggest that cinnamon oil, rich in trans-cinnamaldehyde, possesses significant antidepressant and anxiolytic effects in mice.

Overall, the evidence indicates that cinnamon oil has promising anxiolytic effects in animal models, highlighting its potential as a natural therapeutic agent for anxiety disorders. However, further research, including clinical trials in humans, is necessary to fully understand its efficacy, safety, and the mechanisms underlying its anxiolytic action.

## 7. Anti-Depressant

Depression is a prevalent and severe illness characterized by persistent sadness, slow decision-making, decreased interest in activities, low self-worth, and disrupted sleep patterns. It significantly impacts individuals’ mental well-being and overall quality of life [114], posing a considerable healthcare burden on society [115]. People with depression often feel despair and lose interest in daily activities and social relationships. The primary approach to managing depression involves a blend of psychotherapy and medication, including tricyclics, selective serotonin reuptake inhibitors (SSRIs), and monoamine oxidase inhibitors (MAOIs). While these antidepressants effectively treat many patients, they may not work for everyone and can have side effects [116], such as emotional blunting [117]. Therefore, the search for new, safer, and more effective drugs to treat depression is a key focus of current research, with natural products being a crucial source of potential treatments [118].

Sohrabi et al. [113] investigated the antidepressant properties of cinnamon oil in mice through both acute and subacute treatment protocols. In the acute phase, mice received varying doses of cinnamon oil via intraperitoneal injections, and their antidepressant-like behavior was assessed using the forced swim test (FST). In the subacute phase, mice were administered different doses of cinnamon oil daily for 14 days, and the effects were evaluated using both the FST and the tail suspension test (TST). The results demonstrated that mice treated with cinnamon oil exhibited a significant decrease in immobility time in both tests after 14 days compared to the control group, indicating notable antidepressant effects.

Ma Tianyu et al. [119] investigated the antidepressant effects of a cinnamon oil solid self-microemulsifying drug delivery system (CO–S-SME) using a chronic unpredictable mild stress (CUMS) model in mice. Over eight weeks, mice were exposed to various stress stimuli and assessed using the sucrose preference test (SPT), forced swim test (FST), and open field test (OFT). The stressed mice exhibited decreased sucrose consumption in the SPT, increased immobility time in the FST, and reduced exploratory behavior and locomotor activity in the OFT, indicating depressive-like behaviors. After treatment with CO–S-SME at different doses, the mice showed significant improvements: increased sucrose preference, decreased immobility time, and enhanced autonomous activities compared to the model group. Additionally, CO–S-SME treatment elevated levels of neurotransmitters such as serotonin (5-HT), dopamine (DA), and norepinephrine (NE) in the brain, reduced plasma corticosterone (CORT) levels, and decreased pro-inflammatory cytokines like IL-6, IL-1β, and TNF-α. The treatment also positively modulated the gut microbiota composition, increasing the diversity and abundance of beneficial bacteria. These findings suggest that CO–S-SME exerts antidepressant effects in CUMS-induced depressive mice, potentially through modulation of neurotransmitter levels, stress hormones, inflammatory responses, and gut microbiota.

In summary, cinnamon oil showed significant antidepressant effects in animal depression models. Cinnamon oil and its novel drug delivery system can improve depressive symptoms, such as increasing sucrose preference, increasing spontaneous activity, and reducing immobility time. These results suggest that cinnamon oil has the potential to be used as a natural antidepressant, but there are no relevant human experimental results, and its specific mechanism of action and clinical application still need further study.

## 8. Antitumor

Cancer is currently one of the leading causes of death worldwide [120]. The disease arises from malignant tissue invading nearby healthy tissue and creating metastases in different organs. Its morbidity and mortality rates are on the rise, presenting a significant risk to human health and a worldwide public health concern [121]. Despite the success of many existing anti-cancer medications in prolonging the lives of patients [122], drug resistance frequently arises during treatment [123]. Hence, it is crucial to explore and produce alternative drugs that are highly effective with minimal side effects to impede cancer progression

### 8.1. Anti-Breast Cancer

Breast cancer is the most common malignant tumor among women globally, with 2.26 million new cases reported, surpassing lung cancer for the first time [124]. While early-stage breast cancer has a better prognosis, mid- and late-stage cases pose significant treatment challenges [120]. In recent years, breast cancer incidence in China has been rising, with an increasing number of younger patients [124]. Current treatments include surgical resection, radiotherapy, chemotherapy, endocrine therapy, and molecular-targeted therapy [125]. Despite multidisciplinary approaches improving survival rates, managing the adverse effects of these treatments remains a significant issue.

Xu Xiqiang et al. [126] developed cinnamon oil-loaded chitosan nanoparticles (CS-CO) to investigate their effects on human breast cancer cells (MDA-MB-231). In vitro experiments demonstrated that CS-CO nanoparticles effectively inhibited the proliferation of these cancer cells. This antiproliferative effect was positively correlated with increased lactate dehydrogenase (LDH) release, indicating a disruption of cell membrane integrity. Wound healing and Transwell assays further revealed that CS-CO nanoparticles dose-dependently inhibited tumor cell invasion and metastasis. The study also found that these nanoparticles increased reactive oxygen species (ROS) levels, decreased superoxide dismutase (SOD) activity, and elevated malondialdehyde (MDA) content in MDA-MB-231 cells, suggesting enhanced oxidative stress. Additionally, CS-CO nanoparticles reduced mitochondrial membrane potential and promoted apoptosis. Overall, CS-CO nanoparticles exhibit significant potential in inhibiting tumor growth and inducing apoptosis, offering a promising novel therapeutic strategy in oncology.

Other studies suggest that natural products like cinnamon oil hold promise in cancer therapy due to their bioactive compounds. For instance, a study by Sadeghi et al. [127] demonstrated that cinnamon oil could induce apoptosis in leukemia cells through the mitochondrial pathway, highlighting its potential in cancer treatment. Moreover, the use of chitosan as a drug delivery system enhances the bioavailability and targeting efficiency of the encapsulated compounds [128]. These findings are consistent with Xu Xiqiang et al.’s research [126], which underscores the therapeutic potential of CS-CO nanoparticles in breast cancer treatment. By leveraging the bioactive properties of natural products and the advanced delivery capabilities of nanoparticles, researchers can develop more effective and less toxic cancer therapies. This approach not only offers a promising alternative to conventional treatments but also paves the way for personalized and targeted cancer therapy.

### 8.2. Anti Lung Cancer

Lung cancer is a complex condition characterized by diverse clinicopathological features and is one of the most prevalent types of cancer [129]. Its high prevalence is a leading cause of cancer-related deaths, with incidence rates expected to rise [130]. Current management strategies for lung cancer include surgery, radiation therapy, chemotherapy, immunotherapy, and a combination of traditional and modern medicinal approaches.

Recent research by Nikita Meghani et al. [131] investigated the anticancer effects of a nanoemulsion composed of vitamin D encapsulated in cinnamon oil on human alveolar lung adenocarcinoma cells (A549). Their study demonstrated that this nanoemulsion effectively inhibits the proliferation of A549 cells. Using the comet assay and cytokinesis-block micronucleus (CBMN) assay, they found that the nanoemulsion induces DNA damage in a dose-dependent manner. This DNA damage is associated with cell cycle arrest in the G0/G1 phase, as shown by flow cytometry analysis. Since the progression through the G0/G1 phase is critical for cell division, arresting cells in this phase can impede tumor growth. By halting the cell cycle, the nanoemulsion may allow for the repair of cellular damage, reduce genetic mutations, and prevent further tumor development. These findings suggest that the vitamin D and cinnamon oil nanoemulsion could serve as a promising therapeutic strategy against lung cancer.

Meghani’s experiments further confirmed that treating A549 cells with cinnamon oil led to an increased expression of pro-apoptotic proteins Bax, caspase-3, and caspase-9, along with a decreased expression of the anti-apoptotic protein Bcl-2. These molecular changes correlated with an increase in apoptotic cell numbers and a reduction in mitochondrial membrane potential, culminating in enhanced apoptosis and inhibition of lung cancer cell growth, migration, and invasion. Supporting these findings, Aftab Alam et al. [9] demonstrated that CO-NE exhibited significant cytotoxic effects on A549 cells. In their study, CO-NE showed an IC_50_ value of 18.05 µg/mL, substantially lower than the IC_50_ of 50.21 µg/mL observed for cinnamon oil alone, as determined by the MTT assay. This indicates that the nanoemulsion formulation enhances the bioavailability and anticancer efficacy of cinnamon oil. The CO-NE not only improved the solubility and stability of cinnamon oil but also facilitated better cellular uptake, leading to increased apoptosis in lung cancer cells. Their results align with Meghani’s observations, reinforcing the potential of cinnamon oil nanoemulsions as a promising therapeutic strategy against lung cancer by effectively inducing apoptosis and inhibiting cancer cell proliferation.

A study by Park et al. [132] explored the anticancer effects of cinnamaldehyde, the primary active component of cinnamon oil, on non-small cell lung cancer (NSCLC) A549 cells. Their study demonstrated that cinnamaldehyde significantly inhibits the proliferation of A549 cells by inducing oxidative stress and promoting apoptosis. Specifically, they found that treatment with cinnamaldehyde, especially when combined with hyperthermia at 43 °C, led to a synergistic decrease in cell viability as measured by MTT assays and increased apoptosis indicated by Annexin V staining. The combination therapy also resulted in cell cycle arrest at the G2/M phase and a reduction in mitochondrial membrane potential. At the molecular level, cinnamaldehyde treatment elevated the generation of reactive oxygen species (ROS), which activated downstream signaling pathways, including the mitogen-activated protein kinases (MAPKs) such as ERK, JNK, and p38. This ROS-mediated activation led to increased expression of pro-apoptotic proteins like cleaved caspase-3 and caspase-9, and decreased expression of anti-apoptotic proteins including Bcl-2, Bcl-xL, and Survivin. The use of N-acetylcysteine (NAC), a ROS scavenger, reversed these effects, confirming that the induction of apoptosis was ROS-dependent. These findings support the idea that components of cinnamon oil, such as cinnamaldehyde, can effectively disrupt cancer cell metabolism and survival pathways. By inducing oxidative stress and triggering apoptosis through the ROS-mediated MAPK pathway, cinnamaldehyde shows potential as a valuable component in lung cancer treatment regimens. This adds to the growing body of evidence suggesting that cinnamon oil and its constituents could be developed as novel therapeutic agents against lung cancer.

Therefore, incorporating cinnamon oil or its active compounds into therapeutic strategies could offer a multifaceted approach to combating lung cancer. By uniting its DNA-damaging properties with its ability to induce oxidative stress and promote apoptosis in cancer cells, cinnamon oil may enhance the effectiveness of existing treatments. This comprehensive strategy holds promise for developing novel anti-cancer therapies that leverage multiple mechanisms to inhibit tumor growth and progression.

### 8.3. Anti-Cervical Cancer

Cervical cancer is a prevalent malignant tumor that significantly impacts women, ranking seventh among all malignant tumors [133]. Research indicates that persistent infection with high-risk human papillomavirus HPV16 and 18 is a key factor in the development of cervical cancer [134]. Currently, cisplatin is a commonly used chemotherapy drug for treating cervical cancer. The platinum in cisplatin binds to DNA, disrupting cell mitosis and triggering apoptosis. However, long-term use of cisplatin is dose-dependent [135] and may lead to DNA damage, genotoxicity [136], and potentially carcinogenic effects [137]. Due to its metabolism primarily in the liver and excretion through the kidneys, cisplatin can accumulate in these organs during treatment, resulting in hepatotoxicity and nephrotoxicity [138]. Hence, combination chemotherapy plays a crucial role in the effective treatment of cervical cancer.

Yonika Arum Larasati [139] investigated the cytotoxic effects of cinnamon oil (CO) on cervical cancer HeLa cells to identify a potent and synergistic co-chemotherapeutic agent when paired with cisplatin. Their study demonstrated that CO exhibited dose-dependent cytotoxicity against HeLa cells, with an IC_50_ value of 250 μg/mL. When combined with cisplatin, CO enhanced the cytotoxic effect, allowing for a reduction in the effective dose of cisplatin while maintaining its anticancer efficacy. Flow cytometry analysis revealed that CO induced cell cycle arrest at the S phase, whereas cisplatin caused arrest at the G1 phase. Notably, the combination of CO and cisplatin led to cell cycle arrest at the G2/M phase, indicating a synergistic disruption of cell cycle progression. This combined treatment also increased the induction of apoptosis compared to either agent alone. These findings suggest that CO can modulate the cell cycle, promote apoptosis, and decrease cell viability in cervical cancer cells, especially when used in conjunction with cisplatin. By enhancing cisplatin’s anticancer effects and potentially reducing the required dosage, CO may help minimize adverse reactions and reduce the risk of drug resistance.

In summary, the study not only broadens the potential uses of cinnamon oil in anti-cancer applications but also provides new ideas and breakthroughs for the study of cancer treatment. It also provides new strategies for exploring the biological mechanism of cinnamon oil’s anti-cancer treatment (Figure 3).

## 9. Other Pharmacological Effects

The peripheral nervous system (PNS) is a highly active system that can receive and transmit signals to the central nervous system (CNS) [140]. Nerve damage can lead to various levels of dysfunction, from mild tingling to complete loss of sensation. The extent of sensory or motor impairment depends on factors like the intensity, duration, and location of the injury [141]. Although these impairments are usually not life-threatening, they impose a certain burden on individuals and society. Sciatic nerve injury is one of the most common neuropathies. In a study conducted by S.F. IZCI et al. [142], rats with sciatic nerve injuries were treated with cinnamon oil, followed by pathological examinations of their tissues. The findings of the study indicate that cinnamon oil has notable effects on rats with sciatic nerve injuries, suggesting potential benefits in treating such conditions. This research provides valuable insights into the potential use of cinnamon oil as a therapeutic intervention for addressing nerve injuries and associated impairments.

Functional dyspepsia (FD) is a prevalent gastroduodenal disease characterized by symptoms such as postprandial satiety, early satiety, epigastric burning, and epigastric pain. These symptoms are closely associated with eating habits [143]; a study conducted by Mehdi Zobeiri [29] found that cinnamon oil soft capsules have a certain therapeutic effect on functional dyspepsia.

## 10. Summary

Cinnamon oil, extracted from various species of the *C*. genus, has been highly valued for its medicinal properties since ancient times, particularly in traditional Chinese medicine. This review provides a comprehensive analysis of its chemical composition, pharmacological actions, and diverse applications, highlighting its potential in medical and industrial fields. By systematically evaluating studies from major scientific databases—including Web of Science, PubMed, and ScienceDirect—we thoroughly examine the therapeutic potential of cinnamon oil. Cinnamon oil exhibits a wide range of pharmacological activities, including antimicrobial, anti-inflammatory, antioxidant, anxiolytic, antitumor, and hypoglycemic effects. It is currently an active ingredient in over 500 patented medicines. The oil has demonstrated effectiveness against various pathogens such as “*S. aureus*”, “*Salmonella*”, and “*E. coli*”. Its antimicrobial mechanisms involve disrupting cell membranes, inhibiting ATPase activity, and preventing biofilm formation, indicating its potential as a natural antimicrobial agent.

Moreover, cinnamon oil’s anti-inflammatory properties are evidenced by its ability to suppress inflammatory markers like vascular cell adhesion molecules and macrophage colony-stimulating factors. It has shown promise in improving metabolic health in diabetic patients by enhancing glucose uptake and insulin sensitivity. Additionally, its antioxidant capacity contributes to its therapeutic effects in mitigating oxidative stress-related diseases. Despite the substantial body of research supporting its medicinal value, further studies are necessary to fully elucidate the mechanisms of action and to ensure the safety and efficacy of cinnamon oil in clinical applications. In the context of increasing antibiotic resistance and the demand for sustainable, effective medical treatments, cinnamon oil represents a promising natural alternative with multifaceted therapeutic potential (Table 2).

## 11. VOSviewer

We used VOSviewer (version 1.6.20) to create a figure highlighting cinnamon oil themes, activities, and sources to enhance the review methodology. Each node represents a keyword, with node size indicating the keyword’s importance within the literature. The lines connecting nodes represent co-occurrence relationships between keywords. Keywords are color-coded based on the clusters they belong to, where keywords in the same cluster are more likely to co-occur in the literature (Figure 4).

## Figures and Tables

**Figure 1 pharmaceuticals-17-01700-f001:**
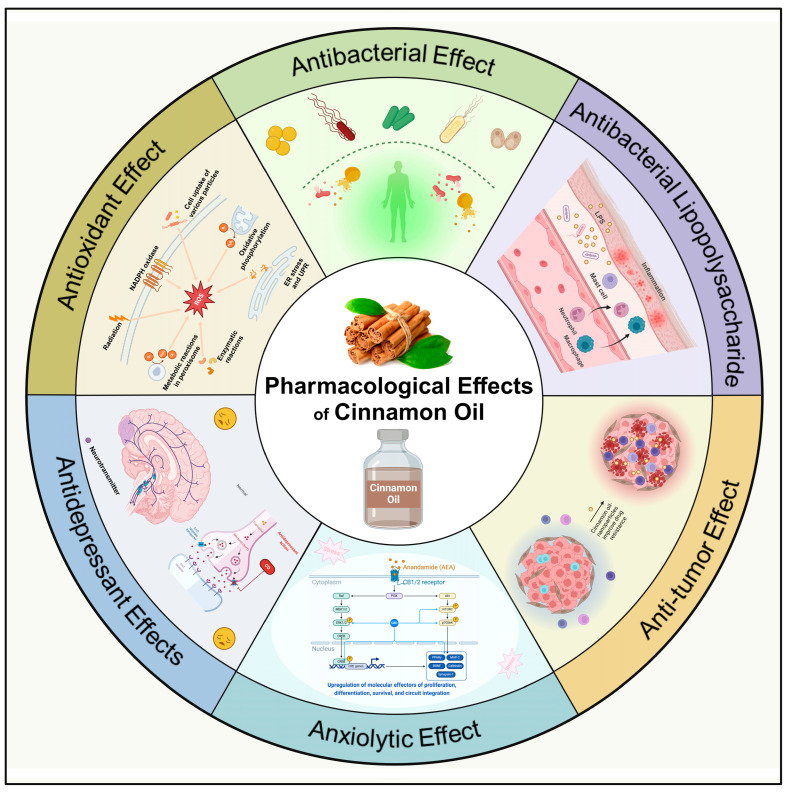
Pharmacological effects of Cinnamon Oil.

**Figure 2 pharmaceuticals-17-01700-f002:**
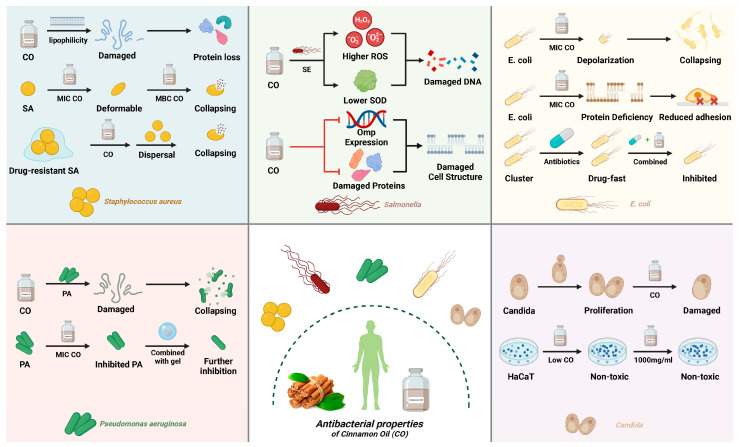
Antibacterial properties of Cinnamon Oil.

**Figure 3 pharmaceuticals-17-01700-f003:**
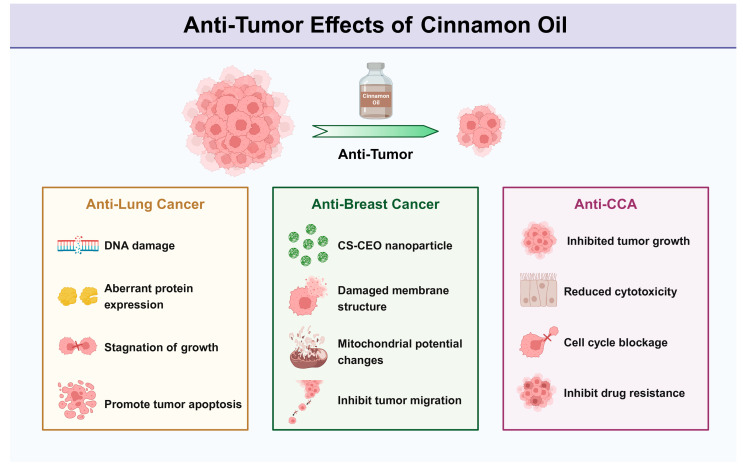
Anti-Tumor Effects of Cinnamon Oil.

**Figure 4 pharmaceuticals-17-01700-f004:**
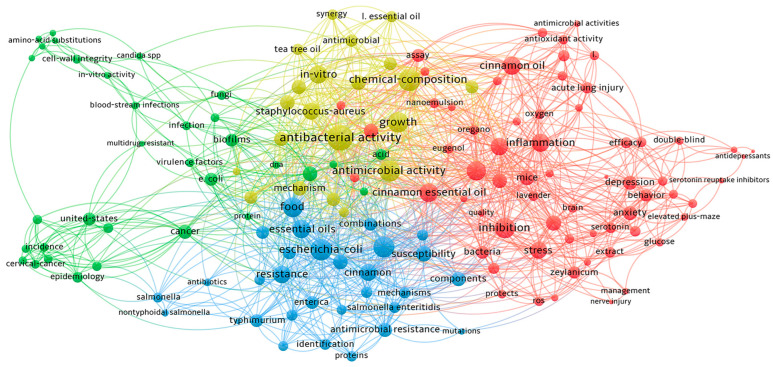
VOSviewer of Cinnamon Oil.

**Table 1 pharmaceuticals-17-01700-t001:** The main chemical components and content ranges of different types of cinnamon essential oils.

Compound	*C. cassia* (%)	*C. verum* (%)	*C. zeylanicum* (%)	*C. burmannii* (%)
Cinnamaldehyde	60–90	60–90	60–90	55–85
Eugenol	5–15	38.42	5–15	10–20
Cinnamyl acetate	5–10	3–8	5–10	5–12
Linalool	1–5	2–6	1–5	2–7
β-Caryophyllene	1–3	1–4	1–3	1–5
Benzyl benzoate	0.5–2	0.5–2	0.5–2	0.5–2
Clovene	0.5–2	0.5–2	0.5–2	0.5–2
Eugenol acetate	0.3–1	0.3–1	0.3–1	0.3–1
Coumarin	0.05–0.5	0.05–0.5	0.05–0.5	0.05–0.5
Inorganic elements	Trace	Trace	Trace	Trace

**Table 2 pharmaceuticals-17-01700-t002:** Pharmacological effects and potential mechanisms of action exhibited by cinnamon oil against different characteristics.

Pharmacological Effects	Aimed Subtype	Main Targets	Potential Mechanisms	Reference
Antibacterial Effect	*S. aureus*	Bacterial cell membrane structure	Components of cinnamon oil can penetrate the lipid bilayer of bacterial cell membranes, altering their physical properties, structure, and function, increasing membrane permeability, leading to leakage of cellular contents and cell death. It can also inhibit ATPase function, disrupt biofilms, and inhibit biofilm formation, exhibiting antibacterial activity against *S. aureus*.	Zhang et al. [46], Franciscato et al. [48], Fadlilah et al. [52], Vasconcelos et al. [53], Wang et al. [54], Shen et al. [55], Gill and Holley [56]
Antibacterial Effect	*S.* spp.	Regulation of oxidative stress	Cinnamon oil induces oxidative stress in *Salmonella* cells by increasing intracellular reactive oxygen species (ROS) levels. Excessive ROS leads to damage of biomacromolecules like proteins and nucleic acids, causing irreversible harm. It decreases the activities of antioxidant enzymes (SOD, CAT, POD), weakening the bacteria’s antioxidant defense. It also increases the permeability of outer membrane proteins, disrupting normal bacterial metabolism.	Zhao et al. [21], Hicks et al. [64], Chen and Yang [65], Vaishampayan and Grohmann [66], Hu et al. [67], Niu et al. [68], An et al. [69], Rocker et al. [70], Futoma-Kołoch et al. [71]
Antibacterial Effect	*E. coli*	Bacterial cell membrane integrity	Cinnamon oil disrupts the permeability and integrity of *E. coli* cell membranes, leading to leakage of electrolytes and macromolecules. It affects the cell membrane potential, causing depolarization, disrupting metabolic activities, and ultimately leading to cell death. When combined with antibiotics, it can achieve synergistic antibacterial effects.	Zhang et al. [46], Casalino et al. [78], Szczepaniak et al. [76], Pereira et al. [79], Atki et al. [80]
Antibacterial Effect	*P. aeruginosa*	Bacterial metabolism and cell structure	Cinnamon oil exhibits broad-spectrum antibacterial effects against *P. aeruginosa*. It disrupts bacterial structure, causing cell morphology collapse and increased release of nucleic acids and proteins due to cell lysis. The minimum inhibitory concentration (MIC) and minimum bactericidal concentration (MBC) are both ≤1.0% (*v*/*v*).	Elcocks et al. [84], Kong et al. [83], Khurshid et al. [85]
Antibacterial Effect	*C. albicans*	Biofilm formation and cell wall integrity	Cinnamon oil inhibits fungal biofilm formation at concentrations below 0.2 mg/mL. It causes significant damage to *C. albicans* cell walls, leading to cell shrinkage and reduced hyphae formation. The minimum fungicidal concentration (MFC) is 2.0 mg/mL, effectively eliminating *C. albicans* within 6 h.	Wijesinghe et al. [90], Veilleux and Grenier [91]
Anti-inflammation	lipopolysaccharide-induced	Regulation of inflammation pathway	Cinnamon oil scavenges free radicals, delays depletion of antioxidant enzymes like SOD and GSH-Px, and reduces malondialdehyde (MDA) levels, thereby mitigating oxidative stress. It decreases TNF-α secretion, inhibits NF-κB pathway activation, and increases Mn-SOD levels, reducing inflammation.	Zhao et al. [5], Liu et al. [12], Hyldgaard et al. [97]
Anti-oxidize Effect	Oxidative cellular damage	Scavenging free radicals	Cinnamon oil exhibits strong free radical scavenging activity, reducing oxidative damage to cells. In DPPH assays, it shows high antioxidant capacity with an EC 50 of 0.03 mg/mL, comparable to positive controls. In human red blood cells, the EC 50 is 0.47 mg/mL, indicating significant antioxidant potential.	Chen et al. [31], Lopez-Cano et al. [100]
Anti-anxiety Effect	All subtypes	Enhance the activity center	Cinnamon oil significantly reduced anxiety behavior in mice through inhalation treatment, as shown by increased open-arm dwell time and decreased closed-arm dwell time. Furthermore, cinnamon oil reversed stress-induced changes in brain electrical activity in a zebrafish model. Cinnamon oil exerts anxiolytic effects by regulating the expression of key genes such as Dcc, Egr2, and Fos.	Nguyen et al. [7], Maximino et al. [111], Sohrabi et al. [113]
Anti-depressant Effect	All subtypes	Modulation of neurotransmitter levels and stress responses	Mice treated with cinnamon oil showed reduced depressive behavior in forced swimming and tail suspension tests, and mice treated with cinnamon oil self-micro emulsifying drug delivery system showed significant reduction in depressive behaviors in behavioral tests after experiencing stress stimulation. improvement. These results suggest that cinnamon oil may become a new natural antidepressant.	Sohrabi et al. [113], Ma et al. [118]
Anti-tumor Effect	Breast cancer	Proliferation and apoptosis of cancer cells	Cinnamon oil chitosan nanoparticles inhibit the proliferation of breast cancer cells by destroying the integrity of cell membranes, inhibiting the metastasis and invasion of tumor cells in a dose-dependent manner, by increasing intracellular ROS levels, reducing SOD activity, increasing MDA content, and reducing mitochondrial membranes. It induces cell apoptosis by electric potential and increases the expression of apoptosis-related proteins.	Xu et al. [125], Sadeghi et al. [126], Parhi [127]
Anti-tumor Effect	Lung cancer	Proliferation and apoptosis of cancer cells	Cinnamon oil damages the DNA of lung adenocarcinoma cells, hinders the cell cycle, and slows tumor cell proliferation. Cinnamon oil regulates the expression of apoptosis-related proteins in tumor cells, reduces the mitochondrial membrane potential of lung cancer cells, induces apoptosis, and inhibits the migration and invasion of tumor cells.	Meghani et al. [130], Alam et al. [8], Park and Baek [131]
Anti-tumor Effect	Cervical cancer	Proliferation and apoptosis of cancer cells	The combined application of cinnamon oil and cisplatin can regulate the cell cycle of cervical cancer HeLa cells, causing the cells to arrest in the S phase, while cisplatin causes the cells to arrest in the G1 phase. The combined application of the two causes the cells to arrest in the G2/M phase, and can also trigger the apoptosis of cervical cancer cells, combined application of cinnamon oil can reduce the occurrence of adverse reactions and drug resistance.	Larasati et al. [138]

## Data Availability

This review article does not involve any original data generation or material collection. All data presented in this article are derived from previously published studies, publicly accessible sources, and reputable databases. Consequently, no additional data or materials are available for sharing.

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
