# Peer review of "Therapeutic Potential of Cinnamon Oil: Chemical Composition, Pharmacological Actions, and Applications"

_pharmaceuticals, 2024, doi:10.3390/ph17121700_

Round 1

Reviewer 1 Report

Comments and Suggestions for Authors

The work has a correct layout, presents the issue of biological activity and use of cinnamon very well. However, the chemical composition is presented very poorly. In the chapter on chemical analysis the use of cinnamon is desribed as well - here it should be removed. The authors should include a table comparing the chemical composition of the essential oil of different species and types of cinnamon. The chemical diversity (content and composition of biologically active compounds) within this genus should be described thoroughly since it influence biological activity of herbal raw materials. There is no information about the chemical composition of the leaves, too (other plant organs than the bark, despite the fact that the authors mention this, were not described). And what about other groups of biologically active compounds apart from essential oils? An important compound here is coumarin (occuring in EO and in the free form), it is worth mentioning the action of such substances. Refferences to the pharmacopoeias requirements should be done, too.

Author Response

Comment 1

The work has a correct layout, presents the issue of biological activity and use of cinnamon very well. However, the chemical composition is presented very poorly. In the chapter on chemical analysis the use of cinnamon is desribed as well - here it should be removed. The authors should include a table comparing the chemical composition of the essential oil of different species and types of cinnamon. The chemical diversity (content and composition of biologically active compounds) within this genus should be described thoroughly since it influence biological activity of herbal raw materials. There is no information about the chemical composition of the leaves, too (other plant organs than the bark, despite the fact that the authors mention this, were not described). And what about other groups of biologically active compounds apart from essential oils? An important compound here is coumarin (occuring in EO and in the free form), it is worth mentioning the action of such substances. Refferences to the pharmacopoeias requirements should be done, too.

Response 1

Thanks for the careful suggestion. 

In response to your suggestions, we have made the following changes to the manuscript to improve the comprehensiveness and scientificity of the chemical composition section: 1. The content about the uses of cinnamon in the chemical analysis section has been removed to ensure that this section focuses on the analysis and description of chemical components. 2. A new table (Table X) has been added to the "Chemical Composition" section to compare the main chemical components and their content ranges of different types of cinnamon essential oils, such as C. cassia, C. verum, C. zeylanicum, and C. burmannii. 3. The description of the differences in the content and composition of bioactive compounds in plants of the genus Cinnamomum has been expanded, and how these differences affect the bioactivity of herbal raw materials has been discussed. 4. A description of the chemical composition of cinnamon leaves, including volatile oils and other bioactive compounds, has been added to the "Chemical Composition" section. 5. The discussion on coumarins in cinnamon has been supplemented, describing their presence in essential oils and free forms and their bioactive roles. 6. References to relevant pharmacopoeia standards such as the Chinese Pharmacopoeia have been added to the "Chemical Composition" section to explain the content requirements of active ingredients in cinnamon essential oil and ensure that the discussion content meets quality and safety regulations. These changes have been highlighted in pink in the Chemical Composition section. We believe that these changes fully respond to the reviewers' comments and improve the scientificity and completeness of the manuscript.

Reviewer 2 Report

Comments and Suggestions for Authors

The current manuscript reviewed the chemical components, pharmacological actions, and potential of cinnamon oil in medical and industrial fields. The article is well written and a fairly comprehensive study has been conducted on the properties and applications of cinnamon essential oil. The results are clearly stated and the diverse uses of this essential oil as a bioactive substance are well explained.

In my opinion, the article can be accepted for publication in the Pharmaceuticals after MINOR corrections as follows:

L24: write the scientific names in italic form.

L48-51, L115-118, …: write the scientific names in italic form. Please do it in all text.

-The origin of cinnamon, the area under cultivation and its production in the world, and China's share of global production should be written in the introduction.

-An explanation should be written about the botanical characteristics of cinnamon.

L76: Write the full names of several pathogenic fungi.

L131: What is the range of Eugenol amount in cinnamon oil?

- Write the main components of cinnamon essential oil, indicating their range of values.

-The genus and species names of plants or microorganisms should be written in full for the first time, and only the first letter of the genus should be written in subsequent times. Follow this throughout the text. For example: change Cinnamomum cassia to C. cassia in L141. Also in L190, … .

- The amounts of essential oil and volatile compounds in cinnamon essential oil also depend on the extraction conditions, and it is necessary to mention the most common method of extracting cinnamon essential oil, which is aqueous extraction (extraction with distilled water).

- Cinnamaldehyde is the main compound in cinnamon essential oil. Write an explanation about its structure and chemical properties.

- In general, is cinnamon essential oil more effective on Gram-positive bacteria or Gram-negative bacteria? Why?

- Do aqueous or alcoholic extracts of cinnamon have the same effects as cinnamon essential oil?

Author Response

We thank the reviewer for their affirmation and detailed comments on the manuscript. Herein we address each reviewer’s comments point by point.

Comment 1

L24: write the scientific names in italic form.

Response 1

Thanks for the careful comment. Based on the reviewer’s comments, we have italicized the scientific names as highlighted in yellow on L24 in the revised manuscript.

Comment 2

L48-51, L115-118, …: write the scientific names in italic form. Please do it in all text.

Response 2

Thanks for the careful suggestion. Based on the reviewer’s comments, we have italicized the scientific names as highlighted in yellow on L48-51, L115-118 and all text.

Comment 3

The origin of cinnamon, the area under cultivation and its production in the world, and China's share of global production should be written in the introduction.

Response 3

We appreciate the useful suggestion. Based on the reviewer’s comments, we have supplemented the origin of cinnamon, the area under cultivation and its production in the world, and China's share of global production with green highlights in the introduction.

Comment 4

An explanation should be written about the botanical characteristics of cinnamon.

Response 4

We appreciate the useful suggestion. Based on the reviewer’s comments, we have supplemented the botanical characteristics of cinnamon with yellow highlights in the introduction.

Comment 5

L76: Write the full names of several pathogenic fungi.

Response 5

Thanks for the careful comment. We have supplemented the full names of several pathogenic fungi with yellow highlights in the revised manuscript.

Comment 6

L131: What is the range of Eugenol amount in cinnamon oil?

Response 6

Thanks for the careful comment. Based on the reviewer’s comments, we have supplemented Eugenol amount in cinnamon oil as highlighted in yellow in the chemical composition section.

Comment 7

Write the main components of cinnamon essential oil, indicating their range of values.

Response 7

Thanks for the careful suggestion. Based on the reviewer’s comments, we have supplemented the main components of cinnamon essential oil and indicated their range of values as highlighted in green in the chemical composition section.

Comment 8

The genus and species names of plants or microorganisms should be written in full for the first time, and only the first letter of the genus should be written in subsequent times. Follow this throughout the text. For example: change Cinnamomum cassia to C. cassia in L141. Also in L190, …

Response 8

We appreciate the useful suggestion. Based on the reviewer’s comments, we have revised the manuscript to ensure that the genus and species names are written in full for the first time, with the genus name abbreviated thereafter.

Comment 9

The amounts of essential oil and volatile compounds in cinnamon essential oil also depend on the extraction conditions, and it is necessary to mention the most common method of extracting cinnamon essential oil, which is aqueous extraction (extraction with distilled water).

Response 9

We appreciate the useful suggestion. Based on the reviewer’s comments, we have supplemented the aqueous extraction method with blue highlights in the chemical composition section of the manuscript.

Comment 10

Cinnamaldehyde is the main compound in cinnamon essential oil. Write an explanation about its structure and chemical properties.

Response 10

Thanks for the careful comment. We have supplemented the structure and chemical properties of Cinnamaldehyde with gray highlights in the chemical composition section of the manuscript.

Comment 11

In general, is cinnamon essential oil more effective on Gram-positive bacteria or Gram-negative bacteria? Why?

Response 11

Thanks for the careful comment. In general, cinnamon essential oil is more effective against Gram-positive bacteria than Gram-negative bacteria. This differential efficacy is primarily due to the distinct structural characteristics of their cell walls.

Gram-positive bacteria possess a thick peptidoglycan layer that is prominently exposed on the outer surface of their cell membrane. This robust and porous structure allows the bioactive compounds in cinnamon oil, particularly cinnamaldehyde, to penetrate more easily. Once inside, cinnamaldehyde disrupts the integrity of the cell wall and membrane, leading to leakage of cellular contents, inhibition of essential enzymes, and ultimately cell death. The high permeability of the peptidoglycan layer in Gram-positive bacteria facilitates the effective action of cinnamon oil's antimicrobial agents.

In contrast, Gram-negative bacteria have a more complex cell wall structure that includes a thin peptidoglycan layer situated between the inner cytoplasmic membrane and an additional outer membrane composed of lipopolysaccharides (LPS). The presence of the LPS layer acts as a formidable barrier, limiting the access of hydrophobic compounds like those found in cinnamon oil. The outer membrane not only impedes the penetration of antimicrobial agents but also contains efflux pumps and porin channels that can expel or restrict the entry of these compounds, enhancing the bacteria's resistance to external threats. Consequently, the protective role of LPS in Gram-negative bacteria reduces the overall susceptibility of these organisms to the antimicrobial effects of cinnamon essential oil.

Comment 12

Do aqueous or alcoholic extracts of cinnamon have the same effects as cinnamon essential oil?

Response 12

Thanks for the careful suggestion. 

Aqueous and alcoholic extracts of cinnamon do not exhibit the same effects as cinnamon essential oil due to differences in their chemical compositions and extraction methods. Cinnamon essential oil, typically obtained through steam distillation or supercritical COâ‚‚ extraction, is rich in volatile compounds like cinnamaldehyde and eugenol, which confer strong antibacterial, anti-inflammatory, antioxidant, and analgesic properties. In contrast, aqueous extracts, produced by boiling cinnamon bark in water, contain more polar compounds such as tannins and flavonoids, offering antioxidant and mild antiseptic benefits but with less potent antimicrobial activity. Alcoholic extracts, made by soaking cinnamon bark in ethanol, include a broader range of bioactive compounds, combining some of the antimicrobial and anti-inflammatory effects of essential oils with enhanced antioxidant properties. However, while alcoholic extracts are more potent than aqueous ones, they still generally do not match the comprehensive antimicrobial efficacy of essential oils. Additionally, essential oils are highly concentrated and require careful handling, whereas aqueous and alcoholic extracts are often safer for internal consumption. Therefore, the choice between these extracts depends on the intended application, with essential oils being preferred for their robust antimicrobial effects and alcoholic or aqueous extracts suitable for broader health benefits with varying degrees of potency.
